# DNA Demethylation Induces Tree Peony Flowering with a Low Deformity Rate Compared to Gibberellin by Inducing *PsFT* Expression under Forcing Culture Conditions

**DOI:** 10.3390/ijms23126632

**Published:** 2022-06-14

**Authors:** Kairong Sun, Yuqian Xue, Zeljana Prijic, Shunli Wang, Tatjana Markovic, Caihuan Tian, Yingying Wang, Jingqi Xue, Xiuxin Zhang

**Affiliations:** 1Key Laboratory of Biology and Genetic Improvement of Horticultural Crops, Institute of Vegetables and Flowers, Chinese Academy of Agricultural Sciences, Ministry of Agriculture and Rural Affairs, Beijing 100081, China; skrskr@foxmail.com (K.S.); xueyuqian2046@163.com (Y.X.); wangshunli@caas.cn (S.W.); tiancaihuan@caas.cn (C.T.); wyingying0909@163.com (Y.W.); 2Institute for Medicinal Plants Research “Dr Josif Pančić”, 11000 Belgrade, Serbia; zprijic@mocbilja.rs (Z.P.); tmarkovic@mocbilja.rs (T.M.)

**Keywords:** *Paeonia suffruticosa*, DNA methylation, photoperiod pathway, McrBC PCR, transcriptome, promoter

## Abstract

Gibberellin (GA) is frequently used in tree peony forcing culture, but inappropriate application often causes flower deformity. Here, 5-azacytidine (5-azaC), an efficient DNA demethylating reagent, induced tree peony flowering with a low deformity rate by rapidly inducing *PsFT* expression, whereas GA treatment affected various flowering pathway genes with strong pleiotropy. The 5-azaC treatment, but not GA, significantly reduced the methylation level in the *PsFT* promoter with the demethylation of five CG contexts in a 369 bp CG-rich region, and eight light-responsive related *cis*-elements were also predicted in this region, accompanied by enhanced leaf photosynthetic efficiency. Through GO analysis, all methylation-closer differentially expressed genes (DEGs) were located in the *thylakoid*, the main site for photosynthesis, and were mainly involved in *response to stimulus* and *single-organism process*, whereas GA-closer DEGs had a wider distribution inside and outside of cells, associated with 12 categories of processes and regulations. We further mapped five candidate DEGs with potential flowering regulation, including three kinases (*SnRK1*, *WAK2,* and *5PTase7*) and two bioactive enzymes (*cytochrome P450* and *SBH1*). In summary, 5-azaC and GA may have individual roles in inducing tree peony flowering, and 5-azaC could be a preferable regulation approach; DNA demethylation is suggested to be more focused on flowering regulation with *PsFT* playing a core role through promoter demethylation. In addition, 5-azaC may partially undertake or replace the light-signal function, combined with other factors, such as *SnRK1*, in regulating flowering. This work provides new ideas for improving tree peony forcing culture technology.

## 1. Introduction

Tree peony (*Paeonia suffruticosa* Andr.), an original Chinese ornamental plant with a history longer than 3000 years, is well-received due to its large and colorful flower as well as high humanistic and economic values [1]. This species has spread across the world in half a century, e.g., in Japan, France, and the USA. Tree peony has a very short flowering period; thus, forcing culture is frequently used to prolong its market supply [2]. As a temperate plant, a certain amount of low temperature for buds is necessary for tree peony re-flowering in the next cycle, whereas under forcing culture conditions, gibberellin (GA) is commonly used to compensate for insufficient bud chilling, but excessive GA treatment often causes flower deformity, which seriously affects its ornamental value. DNA methylation regulates many biological processes, and in tree peony, DNA demethylation was reported to promote bud dormancy release [3]. However, the specific role of DNA methylation in regulating tree peony flowering, especially its interaction with various flowering pathways, remains unclear.

GA, one of the most important plant hormones, is involved in various development processes, such as seed germination, stem elongation, flowering, and fruit development [4,5]. GA includes a large category of compounds and intermediates, and more than 130 GA structures have been identified. Among them, only GA_1_, GA_3_, GA_4_, and GA_7_ are thought to function as bioactive hormones, whereas GA_1_ and GA_4_ are the major active GAs that promote plant growth and development [6]. GA biosynthesis is a complex and multifactor-regulated process, which can be divided into three steps: (i) catalysis of *ent*-kaurene from *trans*-geranylgeranyl diphosphate (GGPP), mainly regulated by copalyl diphosphate synthase (CPS) and *ent*-kaurene synthase (KS); (ii) oxidization of *ent*-kaurene to GA_12_-aldehyde, a general GA precursor, regulated by cytochrome P450 monooxygenases; and (iii) catalysis of GA12-aldehyde into bioactive GAs, regulated by GA20-oxidase (GA20ox) and GA3-oxidase (GA3ox). In this step, the biosynthesis of four functional GAs starts differentiation, the early-13-hydroxylation pathway catalyzes GA_1_ and GA_3_, and the non-13-hydroxylation pathway catalyzes GA_4_ and GA_7_. In addition, the bioactive GAs can be inactivated by GA2-oxidase (GA2ox) as a negative regulator after synthesis [7].

In higher plants, flowering is a complex and precisely regulated process, and a series of flowering-pathway regulatory genes have been identified with strong pleiotropy [8]. Among them, *FLOWERING LOCUS T* (*FT*) and *SUPPRESSOR OF OVEREXPRESSION OF CONSTANS1* (*SOC1*) play major roles in inducing flowering, and both of them are induced by *CONSTANS* (*CO*) under a long-day photoperiod [9,10]. The increased expressions of *FT* and *SOC1* activate the meristem identity genes *APETALA1* (*AP1*) and *LEAFY* (*LFY*), respectively, and then irreversibly induce the turning of the floral meristem as well as flowering [11,12]. In addition, GA regulates *SOC1* expression, but not *FT*, in meristematic tissue in two channels in the GA pathway: one is via the interaction of *GA-INSENSITIVE DWARF1* (*GID1*) and *GA-INSENSITIVE* (*GAI*), two GA signaling genes, and the other is through the inhibition of *SHORT VEGETATIVE PHASE* (*SVP*). In addition, GA also promotes flowering by inducing *LFY* through *SOC1* [13].

In addition to heritable regulation, DNA methylation, a reversible and efficient epigenetic modification, also regulates various biological processes in plants as well as environmental adaption [14,15,16,17,18]. Normally, methylation occurs in the promoter region of a gene, which inhibits its expression by decreasing the transcriptional activity, and vice versa. DNA methylation occurs in three sequence contexts: symmetric CG and CHG as well as asymmetric CHH (H presents A, T, or C). CG and CHG methylation are maintained by METHYLTRANSFERASE1 (MET1) and CHROMOMETHYLASE3 (CMT3), respectively, whereas CHH is maintained by DOMAINS REARRANGED METHYLTRANSFERASE2 (DRM2) [19,20]. DNA methylation influences the flowering process mainly by modulating related gene expression, and in *Arabidopsis*, this process is closely related to flowering time mainly by targeting the *FT* promoter [21,22]. In addition, *TaFT1* in wheat (*Triticum aestivum* L.), *CiLFY* in precocious trifoliate orange (*Poncirus trifoliata*), and *CpSVP* and *CpAP1* in papaya (*Carica papaya* L.) influence the flowering process by gene-specific DNA methylation [23,24,25].

In the forcing culture of tree peony, GA is frequently used to compensate for insufficient bud chilling. We previously found that *PsFT* and *PsSOC1* played key roles in regulating tree peony flowering out of season with the interaction of other genes, such as *PsCO*, involved in the photoperiod, and *PsGID1c* and *PsGAI,* involved in GA pathways [26,27]. In this study, the tree peony ‘Luo Yang Hong’, one of the most applied forcing culture cultivars, was used as a test material. We treated the buds with 5-azacytidine (5-azaC), which is an efficient DNA demethylating reagent, as well as GA and the combination of 5-azaC and GA (A + G) for forcing culture assessment. We observed the morphological changes, measured the GA content, and determined flowering-pathway-related gene expression; we also assessed the DNA methylation level in the promoters of a candidate gene. Furthermore, we systematically analyzed the transcriptome data among different treatments and identified other candidate regulatory genes to explore more potential regulatory patterns or pathways. This work provides more evidence on flowering regulation of tree peony, which may also reveal new ways for improving annual flowering technology.

## 2. Results

### 2.1. Effects of 5-azaC and GA Treatments on the Morphological Changes in Tree Peony

In this study, the bud in the control group developed slightly by 14 d but failed to show color or flowering at 42 d; 5-azaC treatment induced bud development with the petal showing color at 28 d and then finally flowering; GA treatment further accelerated this development, whose bud diameter enlarged to 1.2 and 1.4 times that of the 5-azaC treatment group at 14 and 28 d, respectively, and fully opened at 42 d with a deeper petal color. In the A + G treatment group, the flowering process, as well as petal color and bud diameter, were similar to that in the GA treatment group (Figure 1A–C and Appendix A). We further determined the microscopic changes in the bud in the early stage and observed that both the 5-azaC and GA treatments significantly induced organ differentiation at 9 d, accompanied by a decrease in the starch grain number in both the 5-azaC and A + G treatment groups (Figure 1D–F). The branch length increased significantly in all three treatment groups at 7 and 14 d, but the value in the 5-azaC treatment did not differ from that in the control group at 28 and 42 d (Figure 1G).

### 2.2. Effects of 5-azaC and GA Treatments on Flowering Quality and GA Biosynthesis Regulation

GA supplementation often results in a certain ratio of deformed flowers in tree peony forcing culture. In this study, the 5-azaC treatment resulted in a 67.9% total flowering rate, and 58.9% of the flowers were normal. The GA treatment increased the total flowering rate to 95.4%, but 67.4% of the flowers were deformed at different levels, and 9.3% of them reached Mal III. In the A + G treatment group, the total flowering rate was similar to that observed in the GA treatment, but the normal flower rate was 1.8 times that of the GA treatment group (Figure 2A). In addition, the GA treatment reduced the petal number and area compared to the 5-azaC treatment, whereas the leaf area and chlorophyll content were not affected (Figure 2B–D).

We determined the contents of nine types of GAs in the buds. The GA_1_, GA_3_, GA_29,_ and GA_53_ contents were dramatically induced or reduced by GA treatment, whereas the GA_3_, GA_4_, GA_15_, GA_19,_ and GA_53_ levels were up- or downregulated by 5-azaC treatment. Interestingly, no superposition effect was detected for any of the GAs after A + G treatment (Figure 2E). At the transcriptional level, the expression of *PsKS* was reduced by both the 5-azaC and GA treatment as well as A + G at 14 and 21 d (Figure 2F). *PsGA20ox* expression showed a relatively high level in the control group, but the GA and A + G treatment rapidly reduced this expression from 7 d, followed by the 5-azaC treatment from 14 d (Figure 2G). *PsGA2ox* expression continuously increased in the control group from 7 d, and all three treatments inhibited this increase at a similar level, except the 5-azaC treatment at 21 d (Figure 2H).

### 2.3. Effects of 5-azaC and GA Treatments on the Expression of Flowering-Pathway Genes

We determined the expression of flowering-pathway-related genes in buds, especially those involved in the photoperiod and GA pathways. The results showed that *PsCO* expression was dramatically reduced by GA treatment from 7 d, whereas the reduction by 5-azaC treatment was less intense and started at 14 d (Figure 3A). Both *PsFT* and *PsAP1* were induced by the 5-azaC and GA treatments at 7 and 14 d, with a greater effect on *PsAP1* in the GA treatment group at 7 d (Figure 3B,C). *PsSVP* expression continuously decreased in the control group, whereas the 5-azaC and GA treatments intensified and reduced this decrease, respectively, at 7 d (Figure 3D). *PsSOC1* expression in the control group also decreased continuously, with an intensified effect of the GA treatment at 7 d (Figure 3E). The expression of *PsLFY* showed a single peak curve and peaked at 7 d in all groups, with the lowest level in the GA treatment group (Figure 3F). In the GA pathway, the expression of *PsGID1c* continuously increased in the control group; both the 5-azaC and GA treatments inhibited this increase, with the latter having a greater effect from 7 d (Figure 3G). The 5-azaC treatment had a limited effect on *PsGAI* expression, whereas the GA treatment reduced the expression from 7 d to 21 d (Figure 3H). Interestingly, similar to GA biosynthesis, little superposition effect was detected in all of the gene expressions in the A + G treatment group.

### 2.4. Effects of 5-azaC and GA Treatments on DNA Demethylation-Induced Flowering in Tree Peony

According to our results, the 5-azaC, GA, and A + G treatments induced *PsMET* expression in buds at 7 d but inhibited this increase at 21 d. *PsCMT* expression was also induced by all three treatments but to a lesser extent by the 5-azaC treatment at 7 d. *PsDRM* expression in the 5-azaC and A + G treatment groups remained at a high level at 7 d, and GA treatment reduced it to a similar level as that in the control at 7 d, with a further decrease at 14 d (Figure 4A).

We then measured the DNA methylation levels in the promoters of the eight flowering-pathway genes at 7 d (Appendix A). The results showed that the 5-azaC and GA treatments significantly reduced DNA methylation in the *PsFT* and *PsGAI* promoters, respectively, and GA also slightly reduced DNA methylation in the promoter of *PsLFY* (Figure 4B). We further screened the demethylation sites and identified five CG contexts in a 369 bp CG-rich segment in the *PsFT* promoter whose methylation levels were significantly reduced by the 5-azaC treatment at 7 d (Figure 4C). We predicted 14 *cis*-elements in this 369 bp segment, and interestingly, 8 of them were light-response related, including 4 G-boxes, 3 Box 4, and 1 GT1-motif element (Figure 4D); thus, we further measured the photosynthesis in the leaves of the four groups. Although no obvious morphological differences were observed in the leaves (Figure 4E and Appendix A), both the 5-azaC and GA treatments increased the Pn and Tr levels, whereas the level of Gs was not affected by either treatment. In addition, both treatments decreased the Ci level with a stronger effect by the 5-azaC treatment (Figure 4F).

### 2.5. Transcriptome Data Analysis

To explore more potential regulatory factors or patterns during the flowering process, we performed transcriptome sequencing and obtained 19,746 genes. Among them, 15,889 genes were shared among all four groups with each group having a very similar gene quantity (Appendix A). The Pearson correlation heatmap of gene expression showed that the 5-azaC treatment had the highest ecoefficiency compared with the control, followed by the GA and A + G treatments (Figure 5A). We also compared the DEG patterns through pairwise comparison in different groups. The results showed that the 5-azaC treatment group had more upregulated genes than that of downregulated ones compared to other three groups, respectively, whereas the GA treatment had the opposite pattern to 5-azaC, which had more downregulated genes compared to other three treatments (Figure 5B). We further analyzed the DEG distribution in the control vs. 5-azaC, control vs. GA, and control vs. A + G treatment group. We obtained 235, 822, and 1222 DEGs, respectively. Among them, a total of 52 DEGs, including the overlap of the control vs. 5-azaC and control vs. A + G, but excluding the control vs. GA, were identified and were considered to be methylation-closer DEGs; by the same strategy, another 302 GA-closer DEGs were also identified (Figure 5C and Appendix A).

For the function category, we mapped the methylation- and GA-closer DEGs to the *Arabidopsis* GO enrichment pathway trees of the cellular component and biological process, respectively. In the cellular component, all methylation-closer DEGs were located in the thylakoid, whereas GA-closer DEGs were distributed in the cell junction, membrane, cell, and extracellular region (Figure 5D–F). Concerning the biological process, the methylation-closer DEGs were focused on the response to stimulus and single-organism process, whereas GA-closer DEGs were distributed more extensively and complexly with a total of 12 categories, including various processes and regulations as well as two categories related to methylation-closer DEGs, and among them, growth had the highest fold change compared to *Arabidopsis* (Appendix A and Figure 5G).

### 2.6. Correlation of Different DEGs and Identification of Potential Regulatory Genes

We constructed heatmaps of the methylation- and GA-closer DEGs according to their expression patterns. The methylation-closer DEGs had greater diversity than the GA-closer DEGs. In addition, about two-thirds of the methylation-closer DEGs were upregulated by 5-azaC treatment, and a similar proportion of GA-closer DEGs was downregulated by GA treatment (Figure 6A,B). We further chose nine and six genes with larger differences from the respective methylation- and GA-closer DEGs according to their heatmaps. Through function prediction, we mapped five candidate genes with potential flowering regulation, including three kinases, *SNF1-related protein kinase1* (*SnRK1*), *wall-associated receptor kinase2* (*WAK2*), and *Type IV inositol polyphosphate 5-phosphatase7* (*5PTase7*), and two bioactive enzymes, *cytochrome P450* and *Sphingoid base hydroxylase1* (*SBH1*). The first four genes are methylation-closer DEGs and the last one is a GA-closer DEG. We then revalidated their expression by RT-qPCR and found that all four methylation-closer DEGs were reduced by both the 5-azaC and GA treatments at different levels, whereas the expression of *SBH1* was induced and reduced by 5-azaC and GA treatment, respectively (Figure 6C).

### 2.7. Construction of the Potential Network Involved in Tree Peony Flowering Regulation

Based on these results, we constructed a potential network of DNA demethylation and GA-induced tree peony flowering. In this network, 5-azaC treatment rapidly induced *PsFT* expression by reducing its methylation level in the promoter region, thus initiating the flowering process. In addition, eight light-responsive related *cis*-elements were identified in the *PsFT* promoter, and all methylation-closer DEGs were located in the thylakoid, which is the main site of photosynthesis. Based on these results and the fact that the 5-azaC treatment increased the photosynthetic efficiency of tree peony leaves, we suggest that 5-azaC may partially undertake or replace the function of the light signal, which regulates other kinases and bioactive enzymes, such as *SnRK1* and *cytochrome P450*, to induce tree peony flowering with good quality under forcing culture conditions. By contrast, GA treatment regulated a large number of flowering-pathway genes with strong pleiotropy, and GA-closer DEGs were also widely distributed in- and outside the cells and were associates with various processes and stimuli, which may result in flowers with a high level of deformity. In addition, the 5-azaC and GA treatments may induce tree peony flowering with limited synergism since no superposition effect on GA production or flowering-pathway gene expression was detected in the A + G treatment group, and limited overlap of GO tree categories was found between methylation- and GA-closer DEGs (Figure 6D).

## 3. Discussion

In plants, DNA methylation negatively regulates the flowering process, which is mainly associated with the photoperiod pathway. It was reported that 5-azaC induced flower opening in the long-day plant *Silene armeria* and short-day plant *Perilla frutescens* under non-inductive photoperiodic conditions [28]. Similarly, GA positively affects the flowering time in a range of species, and in *Arabidopsis*, GA is essential for floral induction under unsuitable photoperiods [29]. In our study, both the 5-azaC and GA treatments induced tree peony flowering, with a limited synergistic effect of the A + G group, indicating that DNA demethylation and GA may have individual roles in inducing tree peony flowering (Figure 1). In addition, a moderate GA concentration is very important for its function. We earlier observed that inappropriate GA supplementation readily caused tree peony flower deformity (data not shown). In the current results, GA treatment resulted in a higher rate of deformity than the 5-azaC treatment, which further affected the ornamental value partially due to the reduced petal number and area (Figure 2A,B). Thus, we suggest DNA demethylation could be a preferable regulation approach for tree peony forcing culture, although the flowering rate still needs to be improved.

GA regulation is associated with numerous and diverse intermediates as well as bioactive GAs. In *Arabidopsis*, GA_4_ is physiologically more active than GA_1_ and is more involved in the flowering process. In rice, GA_1_ and GA_4_ dominate during vegetative and flowering development, respectively [4], whereas in tree peony, GA_3_ is superior to GA_4_ in promoting bud endodormancy release under forcing culture conditions [30]. In *Arabidopsis*, overexpression of *GA20ox* induced GA production, whereas GA_3_ applied to a GA-deficient mutant increased *GA2ox1* and *GA2ox2* expression but decreased *GA20ox2* and *GA3ox1* expression, indicating both positive and negative feedback occurred in GA biosynthesis [4]. In addition, *GA2ox2* also negatively regulated seed germination and root growth in response to environmental factors [31]. In our results, GA treatment induced the production of GA_1_, GA_3_, and GA_4_, which also rapidly inhibited the expression of *PsGA20ox* and *PsGA2ox*, whereas the 5-azaC treatment only induced GA_3_ and GA_4_ production together with a faster inhibition of *PsGA2ox* expression (Figure 2E–H). These results indicated that DNA demethylation may be more focused on flowering regulation, accompanied by the deactivation of synthesized GAs by *PsGA2ox*. Interestingly, GA_3_ is normally metabolized much lower than GA_1_ and GA_4_; thus, endogenous GA_3_ content is accumulated at very low levels and rarely detected in plants [32], but in our results, GA treatment dramatically induced the GA_3_ production in buds (Figure 2E). Since we applied GA_3_ for GA treatment (as well as A + G), the inducing of GA_3_ may be partially due to exogenous GA_3_, which should be carefully considered in our future work.

For most perennial temperate plants, flowering at the appropriate seasonal time is crucial for their annual cycling. During this process, various signals are integrated and function in multiple intersecting pathways with pleiotropic gene regulation. In aspen (*Populus trichocarpa*), the flowering timing is controlled by a *CO/FT* regulon in the photoperiod pathway, and *PtCO2* induces *PtFT1*, which then results in flower opening [33]. In chrysanthemum, GA signaling predominates long-day flowering with the up- and downregulation of *GID1* and *GAI*, respectively, thereby inducing the expression of *SOC1* and *LFY* [34]. In grapefruit (*Citrus paradist* Macf.), overexpression of *CsAP1* significantly shortens the flowering time, and *CsLFY* overexpression further shortens this period [35]. In our results, all of the detected flowering-pathway genes were rapidly induced or reduced by GA treatment at 7 d, including *PsFT*, which was not considered to be involved in the GA pathway [13] (Figure 3), indicating that GA has multiple and probably overlapping functions in regulating tree peony flowering.

In plants, 5-azaC inhibits DNA methyl transferase activity, and demethylation occurs globally on the genome after cell divisions [18]. Especially, DNA methylation regulated flower opening by affecting the related gene expression via demethylating the related promoter region. In *Arabidopsis*, DNA methylation in the *FT* promoter reduced its expression, and as a result, delayed the flowering process [22]. In wheat, disruption of DNA methylation in the *TaFT1* promoter also changed its flowering time [23]. In papaya, early flowering was associated with the gene-specific demethylation of *CpSVP* and *CpAP1* [25]. In potato (*Solanum tuberosum* L.), DNA demethylation induced photoperiod- and GA-pathway-related gene expression and then promoted tube initiation, which was similar to flowering regulation [36]. In our results, among the eight flowering-pathway-related genes, only *PsFT* was rapidly and intensely induced by 5-azaC treatment at 7 d, and its promoter was also significantly demethylated at 7 d with five related CG contexts being identified (Figure 3 and Figure 4). These findings indicate that *PsFT* may play a core role in DNA demethylation-induced tree peony flowering. In addition, we identified eight light-response-related *cis*-elements in the tested *PsFT* promoter segment, and 5-azaC enhanced the photosynthetic efficiency by increasing Pn and Tr in leaves (Figure 4D,F). In addition, GO enrichment analysis of cellular components indicated that all the Me-closer DEGs were located in the thylakoid, which is the main site for photosynthetic reactions. These results further indicated that 5-azaC may partially function as light or photoperiodic signal to induce *PsFT* expression as well as the flowering process.

In addition to flowering-pathway genes, DNA methylation also modulated numerous other genes with spatiotemporal expression during flowering regulation. In *Arabidopsis*, DNA methylation regulates flower opening in a complex manner, which has been correlated with more than 3000 DEGs [18]. In rice, DNA demethylation is critical for its reproductive development, and in a DNA hypermethylation mutant, 3898 downregulated and 2716 upregulated genes were identified [37]. In tree peony, 5-azaC treatment accelerated bud dormancy release, and more than 1800 DEGs were identified during this process [38]. In addition, DNA methylation regulates plant development in distinct ways among different genotypes, tissues, and other factors. In potato, photoperiod-sensitive and -insensitive genotypes had their own independent DNA methylation modes and shared a few differentially methylated genes [36]. In *Taihangia rupestris*, the DNA methylation patterns varied in staminate and perfect flowers [39]. According to our results, among the four groups, the 5-azaC and GA treatments had the highest number of up- and downregulated genes, respectively (Figure 5B). We also found that the GO cellular component and biological process categories shared no or very few DEGs between the 5-azaC and GA treatments (Figure 5F,G), thus providing more evidence for our suggestion that the mechanism through which these two treatments regulate tree peony flowering may vary.

According to the hierarchical clustering analysis of methylation- and GA-closer DEGs, we further mapped five potential flowering regulatory genes. Among them, *SnRK1* is involved in sugar signaling for negative regulation of tree peony flowering as we reported previously [40]; *WAK2* is a serine/threonine-protein kinase and may function in regulating cell expansion, morphogenesis, and development [41]; *5PTase*7, a member of a large family of 5Ptase enzymes, acts in coordinating plant responses to different stresses [42]; *cytochrome P450* is involved in many processes of plant primary metabolites, such as GA biosynthesis [43]; and *SBH1* regulates the transformation from vegetative to reproductive stages, and the loss-of-function mutant fails to complete this transformation, accompanied by the induced expression of programmed cell death-associated genes [44]. These findings indicate that tree peony flowering regulation is a complex and multifactor-regulated process and can provide more ideas for supplementing our potential network (Figure 6D).

## 4. Conclusions

In this study, 5-azaC treatment regulated tree peony flowering with a low deformity rate by rapid induction of *PsFT* expression through a reduction in its methylation level in the promoter region, and eight light-responsive related *cis*-elements were predicted in a 369 bp CG-rich segment of this promoter. In addition, 5-azaC treatment enhanced the photosynthetic efficiency in leaves, and all of the methylation-closer DEGs were located in thylakoid, the main site for photosynthesis. Thus, we suggested that 5-azaC may partially undertake or replace the function of the light signal to induce tree peony flowering. Furthermore, we constructed a potential regulatory network for tree peony flowering, which may provide some new ideas for improving tree peony forcing culture technology.

## 5. Experimental Procedures

### 5.1. Plant Materials, Growth Conditions, and Sample Collection

In this study, the 5-year-old adult tree peony (*P. suffruticosa* Andr.) ‘Luo Yang Hong’ was used as a material, which was grown in the experimental field of the Chinese Academy of Agricultural Sciences, Beijing, China. All of the plants were placed in plastic pots (40 cm × 40 cm) immediately after being dug from the ground in mid-September and were then transferred into a greenhouse in early December for forcing culture. The experiment was repeated for 2 years, and in each year, a total of 100 plants with similar growth and health conditions were randomly selected and divided into four groups (25 plants for each group) for further use.

(1)Control group: the plants were placed in a regular forcing culture environment with conventional water and fertilizer management according to [45]. The environmental condition is shown in Appendix A.(2)5-azaC treatment group: all buds were supplied with 5-azaC (100 μmol L^−1^, biological reagent grade, Shanghai Yuanye Bio-Technology Co., Ltd., Shanghai, China) 14, 16, and 18 d after the plants were transferred into the greenhouse.(3)GA treatment group: all buds were supplied with GA_3_ (600 mg L^−1^, analytical grade, Beijing Chemical Reagent Co., Ltd., Beijing, China) at the same time as the 5-azaC treatment group.(4)A + G treatment group: all buds were supplied with 5-azaC and GA_3_ at the same concentrations and times described above, and the processing interval was 2 h with 5-azaC as the first treatment.

All of the treatments were applied in the morning of a sunny day with the same management and environment as those in the control group. The first day of 5-azaC treatment was set as 0 d.

### 5.2. Observation of Morphological Changes

Ten buds from each group (one bud per plant) were randomly selected for the flowering process, which was recorded photographically at 0, 7, 14, 28, and 42 d, and all buds were used for flowering quality assessment at 28 d based on the following standards: (1) No flower: the bud stops to enlarge with no color showing; (2) Normal: the outer petals are tightly wrapped with the normal color showing; (3) Mal I: the top part of the outer petals is slightly expanded with a few anthers showing among the petals; (4) Mal II: the outer petals are half expanded with an irregular shape, more anthers can be seen, and the stigmas are still beneath the petals; (5) Mal III: most of the petals are expanded with an irregular shape, and both anthers and stigmas are almost completely exposed.

The histological observation of buds was through a Leica Microsystems DM5500B microscope (Wetzlar, Germany), and the micro-sections of the buds were prepared using a rotary microtome (RM2255, Leica, Germany). The Cold Field-Emission Scanning Electron Microscope (SEM) observation of starch grains was through a Hitachi SU-8010 SEM (Tokyo, Japan), and the sample preparation was according to the standard protocols. The *CIE-LAB* color parameters were measured using a Konica Minolta chromameter (CR-400; Minolta Co., Ltd., Osaka, Japan). The color parameters l (100 means white, 0 means black), a (redness (+) and greenness (−)_, and b (yellowness (+) and blueness (−)) were determined in the CIE Lab space under the illuminant D65/10 °C. ΔE indicates the total color difference.

The number of starch grains was calculated by ImageJ software (1.52a, National Institutes of Health, Bethesda, MD, USA), and the petal and leaf areas were measured using Plant Image Analyzer software (LA-S, Hangzhou Wanshen Testing Technology Co., Ltd., Hangzhou, China).

### 5.3. Measurement of Chlorophyll, GA Contents, and Photosynthetic Gas Exchange

For chlorophyll content measurement, the third leaf from the top of the plant was measured using a portable chlorophyll meter (TYS-B, Zhejiang Top Cloud-agri Technology Co., Ltd., Hangzhou, China) at 28 d. For GA content measurement, the pestle sample of buds without bracts at 14 d was dissolved in ACN/H_2_O (90:10, *v*/*v*), and an internal standard mixed solution was added. After centrifugation and the addition of triethylamine and 3-bromopropyltrimethylammonium bromide, the supernatant was incubated at 90 °C for 1 h and then evaporated to dryness. Then, the supernatant was redissolved in ACN/H_2_O and filtered for further UPLC-ESI-MS/MS (UPLC, ExionLC™ AD; MS, Applied Biosystems 6500 Triple Quadrupole) analysis based on the manufacturer’s instructions, and GA contents were detected by MetWare based on the AB Sciex QTRAP 6500 LC-MS/MS platform (Shanghai Ab Sciex Analytical Instrument Trading Co., Ltd., Shanghai, China).

Photosynthetic gas exchange was measured using the third leaf each plant at 28 d from 9:00 to 11:00 am; the net photosynthetic rate (Pn), stomatic conductance (Gs), intercellular CO_2_ concentration (Ci), and transpiration rate (Tr) were measured using a CIRAS-3 portable photosynthesis system (PP Systems, Amesbury, MA, USA) under ambient CO_2_ concentrations (380–420 μmol·mol^−1^, chemicals removed) at 25 °C, 1200 μmol (photon) m^−2^ s^−1^ light (90% red light and 10% blue light), and 80% relative humidity.

### 5.4. RNA Isolation and RT-qPCR Testing

The total RNA was isolated from flower buds using an RNA prep Pure Plant Kit (Aidlab, Beijing, China), and its quality and quantify were assessed by 1.2% agarose electrophoresis and NanoDrop 2000c spectrophotometer (Thermo Scientific, Waltham, MA, USA), respectively. The primers for RT-qPCR were designed using Primer 5.0 software. The relative gene expression was calculated by a double standard curve according to the CFX96 Real-Time system (Bio-Rad, Irvine, CA, USA). *Actin* was used as the reference gene, and the 2^−ΔΔCt^ method was used for relative gene expression data analysis [46]. The primer and gene information are shown in Appendix A.

### 5.5. Measurement of the DNA Methylation Level

For McrBC-PCR, the genomic DNA extracted from buds at 7 d was digested with McrBC (New England Biolabs Inc., Ipswich, MA, USA) according to the manufacturer’s recommendations for 4 h at 37 °C; the same reaction without GTP was used as a negative control. PCR was performed using 20 ng McrBC-treated DNA. The thermal cycling conditions were: 95 °C for 5 min followed by 28–30 cycles of 30 s at 95 °C, 30 s at 55–61 °C and 70 s at 72 °C, and finally 10 min at 72 °C. The primer sequences are listed in Appendix A.

For MassARRAY analysis, bisulfite conversion of genomic DNA from the buds at 7 d was performed using the EZ DNA Methylation Kit (Zymo Research, Irvine, CA, USA) according to the instruction manual. Quantitative methylation analysis of bisulfite-treated genomic DNA was conducted through Agena’s MassARRAY EpiTYPER system (Agena Bioscience, San Diego, CA, USA). The mass spectra were obtained by MassARRAY MALDI-TOF-MS and analyzed by EpiTYPER^TM^ software. Duplicate independent analyses of each bisulfite-treated sample were performed. Data of poor quality for the quantitative methylation detection of each CG site were excluded. The primer sequences are listed in Appendix A.

### 5.6. Data Analysis and cis-Element Prediction

All of the experimental data were analyzed using SAS 9.4 (SAS Inc., Cary, NC, USA) and OriginPro 2019b (OriginLab Software Inc., Northampton, MA, USA) software. Differences in morphological, physiological, and gene expression characteristics were analyzed using one-way ANOVA with post hoc Duncan’s tests. *p* values ≤0.05 were considered to be statistically significant. The *cis*-elements on the gene promoter were predicted by PlantCARE [47].

### 5.7. Transcriptome Sequencing and Bioinformatics Analysis

Total RNA was extracted from the buds at 14 d and purified to mRNA by Oligo (dT)-attached magnetic beads. Single-stranded circular DNA was formatted as the final library, which was sequenced, generating 100 bp paired-end reads on the BGIseq500 platform (BGI-Shenzhen, China). The raw transcriptome data were deposited in the NGDC database (CRA006504, accessed on 31 March 2022, https://ngdc.cncb.ac.cn). The sequencing data were filtered with SOAPnuke (v1.5.2), and clean reads were mapped to the reference genome using HISAT2 (v2.0.4) [48]. Pearson’s correlations of all gene expression levels between different groups were analyzed by a heatmap according to their coefficients. The up- or downregulation of different genes between groups was presented using volcanic scatter plots. The crossing and overlapping of differentially expressed genes (DEGs) between different groups were shown in a Venn diagram. The Gene Ontology (GO) enrichment pathways of tree peony based on the gene function in *Arabidopsis* were constructed through agriGOv2 [49], and the fold change in different terms = (number of DEGs with a specific function in tree peony/number of total DEGs analyzed in tree peony)/(number of genes with a specific function in *Arabidopsis*/number of total genes in *Arabidopsis*). The heatmaps of selected DEGs were drawn by Pheatmap (v1.0.8), and the differential expression analysis was performed using PossionDis with a false discovery rate (FDR) ≤0.001 and |Log2Ratio| ≥1. The revalidation of candidate gene expression was performed using RT-qPCR, and the primer sequences are listed in Appendix A.

## Figures and Tables

**Figure 1 ijms-23-06632-f001:**
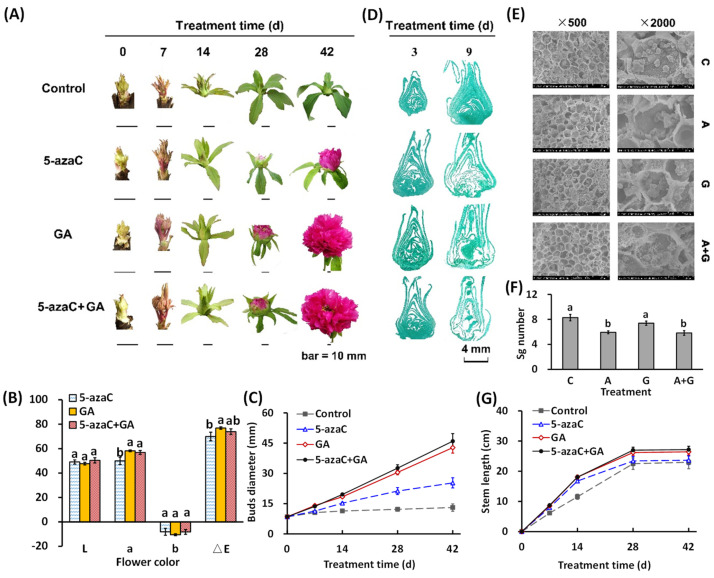
Effects of 5-azaC and GA treatments on the morphological changes in tree peony. (**A**) Morphological changes of buds during the flowering process. (**B**,**C**) Statistical analysis of the petal CIE-LAB color and bud diameter. For color parameters, L: 100 means white, 0 means black; a, redness (+) and greenness (−), and b, yellowness (+) and blueness (−). (**D**) Microstructure observation of buds. (**E**) Electron microscopic observation of buds. (**F**,**G**) Statistical analysis of the starch grain number and branch length. (All data were analyzed by Duncan’s test at *p <* 0.05 after analysis of variance; data are shown as the mean ± SD, for (**B**,**C**) and (**G**), *n* = 10; for (**F**), *n* = 5). note: C, A, and G represent the control, 5-azaC, and GA treatment, respectively.

**Figure 2 ijms-23-06632-f002:**
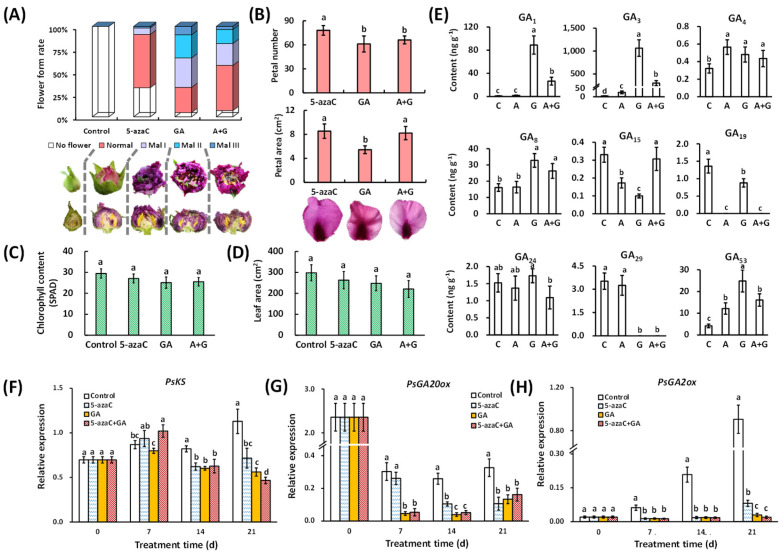
Effects of 5-azaC and GA treatments on flower quality and GA biosynthesis regulation. (**A**) The ratio of normal and malformed flowers in different treatments. The meaning of *Mal I* to *III* can be found in Section 5.2. (**B**) Statistical analysis of the petal number and area. (**C**,**D**) Statistics analysis of the chlorophyll content and leaf area. (**E**) GA content in buds at 14 d. (**F**,**H**) Relative expression of GA biosynthesis-related genes. (All of the data were analyzed by Duncan’s test at *p*
*<* 0.05 after analysis of variance; data are shown as the mean ± SD, and the different letters indicates significant differences among different treatments, for (**B**,**D**), *n* = 10; for (**C**), *n* = 5; for (**E**–**H**), *n* = 3). note: C, A, and G represent the control, 5-azaC, and GA treatment, respectively.

**Figure 3 ijms-23-06632-f003:**
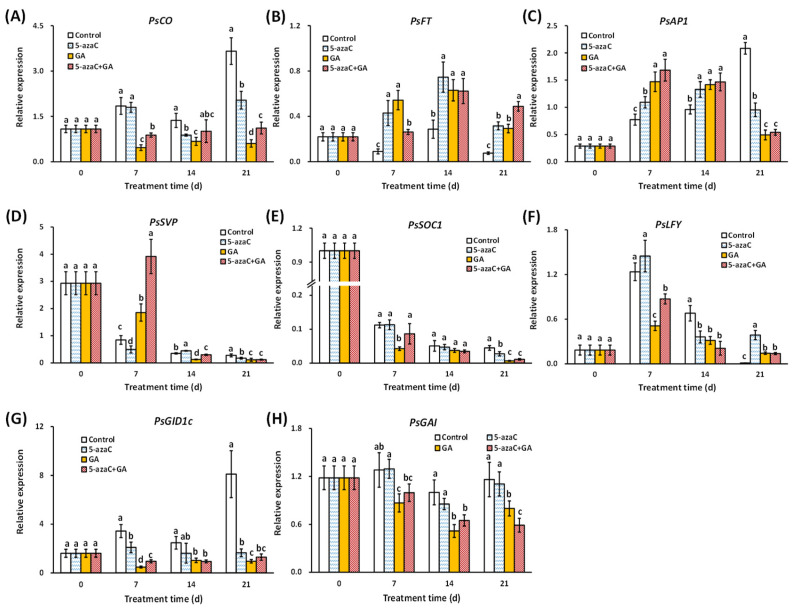
Effects of the 5-azaC and GA treatments on the expression of flowering-pathway-related genes in buds determined using RT-qPCR. (**A**) *PsCO*. (**B**) *PsFT*. (**C**) *PsAP1*. (**D**) *PsSVP*. (**E**) *PsSOC1*. (**F**) *PsLFY*. (**G**) *PsGID1c*. (**H**) *PsGAI*. All experiments were performed with at least three biological replicates, and all expression levels were normalized to the *Actin* control. Error bars indicate the ±SD, and different letters for each day indicate significant differences at *p*
*<* 0.05 (Duncan’s tests).

**Figure 4 ijms-23-06632-f004:**
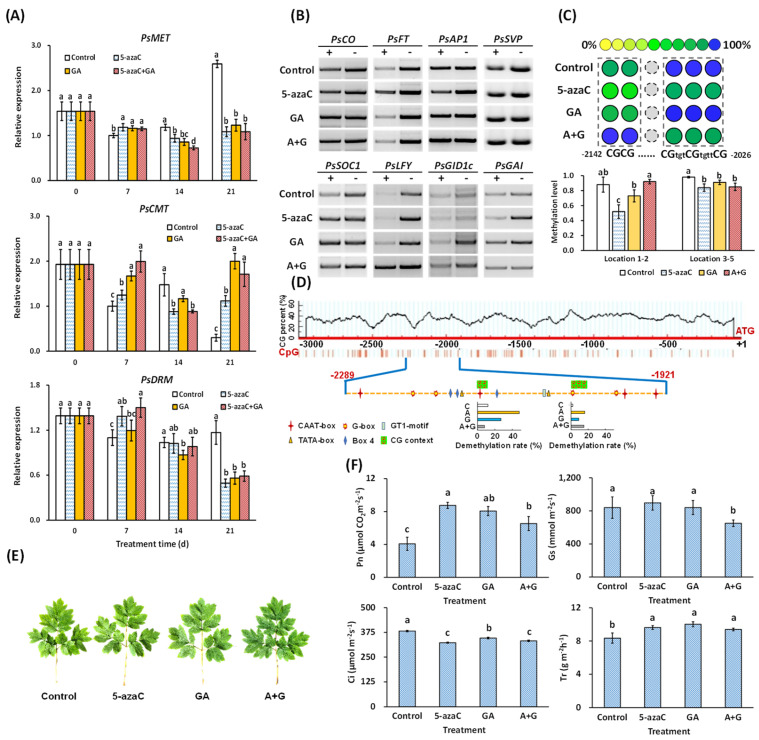
Effect of 5-azaC and GA treatments on DNA methylation levels in the promoter regions of flowering-pathway-related genes and photosynthetic analysis. (**A**) Related expression of DNA methylation regulation genes *PsDRM*, *PsCMT,* and *PsMET* by RT-qPCR (Duncan’s test at *p*
*<* 0.05, error bars indicate the ±SD and different letters for each day indicate significant differences, *n* = 3). (**B**) DNA methylation level in the promoter of flowering-pathway-related genes determined by McrBC-PCR; + and − indicate *with* and *without* GTP, respectively, in the digestion solution (*n* = 3, representative results shown here). (**C**) DNA methylation in the promoter of *PsFT* based on MassARRAY (Duncan’s test at *p*
*<* 0.05, error bars indicate the ±SD and *n* = 3). (**D**) CG context prediction 3 kb before the ATG site in the *PsFT* promoter and the *cis*-element prediction in the tested region. The demethylation level in the above CGs 1–2 and 3–5 are also shown here. (**E**) Morphological levels used for photosynthetic determination. (**F**) Analysis of net photosynthetic rate (Pn), stomatic conductance (Gs), intercellular CO_2_ concentration (Ci), and transpiration rate (Tr) in the third leaf of each plant at 28 d. (Duncan’s test at *p* < 0.05 after analysis of variance; data are shown as the mean ±SD, and the different letters indicates significant differences among different treatments, *n* = 10).

**Figure 5 ijms-23-06632-f005:**
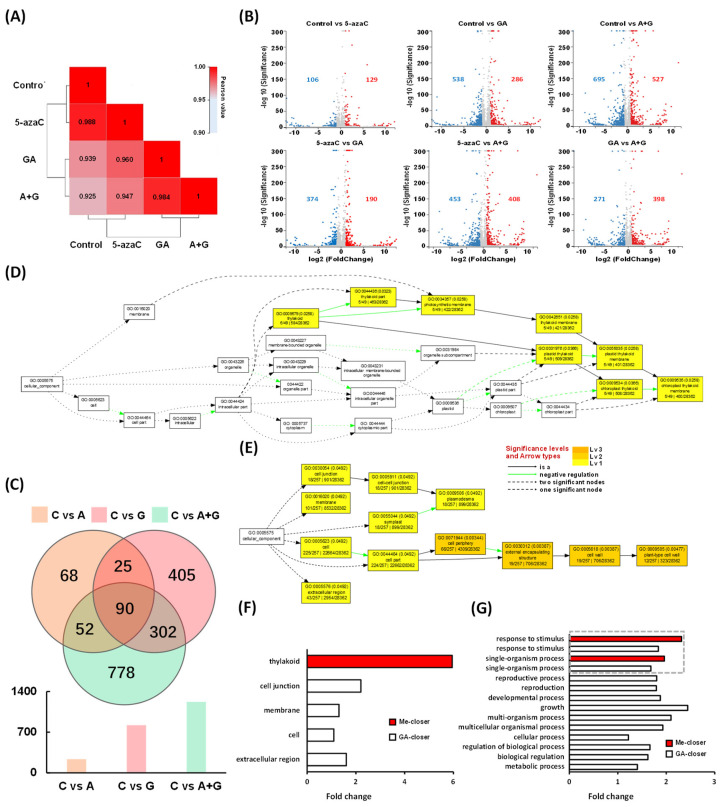
Transcriptome data analysis of differentially expressed genes (DEGs) in buds among different treatments at 14 d. (**A**) Pearson correlation heatmap of all gene expression levels between the treatment groups of each sample according to their coefficients. (**B**) Volcanic scatter plots based on pairwise comparisons of DEGs among different treatment groups; the red and blue plots represent up- and downregulated DEGs, respectively, and the gray plots represent non-DEGs. (**C**) Venn diagram of DEGs in the control vs. 5-azaC, control vs. GA, and control vs. A + G treatment. (**D**,**E**) GO terms of the cellular component in methylation- and GA-closer DEGs, respectively. (**F**,**G**) Quantification of GO terms of the cellular component and biological process in methylation- and GA-closer DEGs, respectively, compared to *Arabidopsis*. note: C, A, and G represent the control, 5-azaC, and GA treatment, respectively.

**Figure 6 ijms-23-06632-f006:**
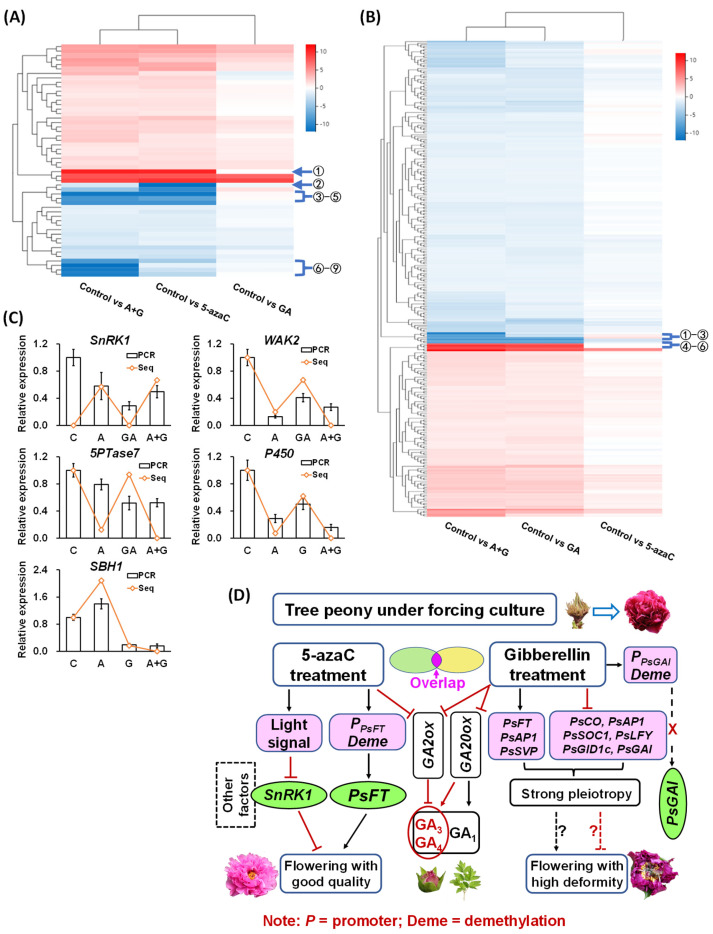
Heatmap of correlations among different DEGs and the potential flowering regulatory network of tree peony. (**A**) Hierarchical clustering analysis of methylation-closer DEGs. The circles indicate candidate DEGs with a greater difference. 1. *SNF1-related protein kinase1* (*SnRK1*); 2. *Violaxanthin de-epoxidase*; 3. Unknown; 4. *G-type lectin S-receptor-like serine/threonine-protein kinase*; 5. *Ubiquitin carboxyl-terminal hydrolase23*; 6. *Cytochrome P450*; 7. Unknown; 8. *Wall-associated receptor kinase2* (*WAK2*); 9. *Type IV inositol polyphosphate 5-phosphatase7* (*5PTase7*). (**B**) Hierarchical clustering analysis of GA-closer DEGs. The circles indicate candidate DEGs with a greater difference. 1. Unknown; 2. Unknown; 3. *Sphingoid base hydroxylase1* (*SBH1*); 4. *β-amylase2*; 5. *Polygalacturonase*; 6. *Rhodanese-like domain-containing protein11*. (**C**) Expression of candidate flowering-regulated genes. PCR represents the RT-qPCR results, all expression levels were normalized to the *Actin* control, and error bars indicate the ±SD (*n* = 3); Seq represents transcriptome sequencing results. (**D**) Potential network construction of 5-azaC- and GA-induced tree peony flowering under forcing culture.

## Data Availability

The data used to support the findings of this study are available from the corresponding authors upon request.

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
