# Peer review of "DNA Demethylation Induces Tree Peony Flowering with a Low Deformity Rate Compared to Gibberellin by Inducing PsFT Expression under Forcing Culture Conditions"

_ijms, 2022, doi:10.3390/ijms23126632_

Round 1

Reviewer 1 Report

The manuscript by Sun et al described that DNA methylation inhibitor promotes tree peony flowering with low flower deformity rate than flower induction by gibberellin. They analyzed the precise mechanism of the phenomenon. They tried many approaches to elucidate the events which occurred after the treatment of DNA methylation inhibitor, gibberellin, and both. However, following concerns raised.

1. In the introduction (Page 2), it was written as ‘In addition, GA regulates SOC1 expression, but not FT, in two channels in the GA pathway,,,’. This description is not accurate. Because GA induction in leaf induces FT expression in leaf, above regulation is only observed in meristematic tissue.

2. It is very important to describe the precise sample information to analyze and to conclude something. In experimental procedures part, the sample information (which tissue and what timing of sample was used) was not mentioned in GA content measurement and in analysis of Pn, Gs, Ci, and Tr. Moreover, experimental procedures part was located very latter in this IJMS paper format. It is difficult to understand and assess the without sample information in results section and figure legends. It should be shown the tissue information for expression analysis of flowering-pathway-related genes (page 5), DNA methylation analysis (page 6), and it should be shown the sampling timing after treatment for transcriptome analysis (page 7).

3. Page 5 and page 11 (and figure 6), it was concluded GA3 was induced after GA treatment. However, GA3 itself was used for GA treatment. It can not be excluded the possibility that the measured GA3 was derived from treated GA3 itself.

4. Page 5, the description ‘Both PsFT and PsAP1 were induced by the 5-azaC and GA treatments’  and ‘PsSOC1 expression in the control group also decreased continuously, with an intensified effect of the GA treatment at 7 d (Fig. 3E).’ was inconsistent with the introduction of ‘In addition, GA regulates SOC1 expression, but not FT, in two channels in the GA pathway,,,’. It should be discussed.

5. 5-azacytidine inhibits DNA methyl transferase activity and demethylation occurs globally on the genome after cell divisions. However, DNA demethylation was detected only FT promoter region after 5-azacytidine treatment in this manuscript. I think it is very rare situation. It should be discuss about this point.    

6. In this manuscript, only each one peony gene was used and named correspond to Arabidopsis genes, and analyzed the expression. It is not shown any genome information, is there any copy genes for each in tree peony? How was decided the each corresponding genes?

7. I figure 4 (B), dots data and bar charts result show different in location 1-2.

8. Page 6, it was mentioned ‘We predicted 14 cis elements in this 369-bp segment, and interestingly, eight of them were light response related’. It was not shown which elements were light responsive.

9. Page 7, it was mentioned ‘The results showed that the 5-azaC treatment group had the highest number of upregulated genes’. However, the number is 129, it is smallest in all comparison.

10. Page 8, all methylation-closer DEGs were located in the thylakoid. There is no discussion about the meaning.

Minor points

1. In all figure, (A), (B), etc. were shown in left side of each figure.

2. In figure 1 (B), explanation was needed for petal CIE-LAB color. What does L, a, b and E mean?

3. In figure 2 (A), explanation was needed for Mal I , II, and III.

4. In Figure4 (F), explanation was needed for Pn, Gs, Ci, and Tr.

5. Page 4, Mal III and PsKS gene were appeared in the manuscript without any explanation before.

Reviewer 2 Report

Review Report

The research article titled ‘DNA demethylation induces tree peony flowering with a low deformity rate compared to gibberellin by inducing PsFT expression under forcing culture conditions’ by Sun et al., is an interesting and extensive work. It presents the regulation of tree peony flowering by 5-azaC with a lesser flower deformity rate in contrast to gibberellin induced regulation. Authors claimed that 5-azaC regulated the rapid induction of PsFT expression through reduced methylation in PsFT’s promoter with the demethylation in a 369bp CG-rich region. The demonstration of individual roles of GA and 5-azaC in inducing the tree peony flowering and the elucidation of potential regulatory network for tree peony flowering is interesting and convincing. Presentation of the article is clear and appropriate for the journal. These findings provides a new scope in peony forcing culture. The manuscript is well-written and well structured. The objectives fit well with the scope of the journal and the materials and methods are appropriately described. Hence, I believe the manuscript merits the publication in International Journal of Molecular Sciences. There are no substantial comments to add. However, the following corrections are required.

1. Please mention the number of plants used per group in the section 5.1 of Experimental Procedures. Is it 10 or 25 per each group?

2. In section 5.3, please mention the CO2 concentration. Please also mention the time at which the measurements were done.

3. Please mention whether RNA is isolated from leaf or flower in section 5.4.

4. Except transcriptome analysis, there is no clear mention of whether the experiments are performed at 14 days or 25 days. Please mention it clearly in each section in Experimental procedures.

5. X-axis in the second graph of Fig. 4(A) is absent. Please check.

6. Why is the demethylation rate of A+G in the first graph of Figure 4(D) less, compared to individual A and G treatments, which should apparently be similar to either A or G?

Round 2

Reviewer 1 Report

The authors have addressed and answered the most major points raised in the previous reviews adequately. I only suggest following point.  

Page 5 and page 11 (and figure 6), it was concluded GA3 was induced after GA treatment. However, GA3 itself was used for GA treatment. It can not be excluded the possibility that the measured GA3 was derived from treated GA3 itself. 

Response: Thank you for your kind comments. The sample was taken after two weeks of the application of GA3. For the sampling collection, the outer leaves were stripped, cleaned, and quick freezing, which should exclude the effect of exogenous gibberellin.

My new comment; It is known that GAis normally metabolized much lower than GA1 and GA4 in plants (e.g. Spray CR, Kobayashi M, Suzuki Y, Phinney BO, Gaskin P, MacMillan J. 1996. The dwarf-1 (dt) mutant of Zea mays blocks three steps in the gibberellin-biosynthetic pathway. PNAS. USA 93:10515–18). Thus endogenous GA3 is accumulated very low amount and rarely detected in plants. I do not agree authors insistence that detected GAin Fig 2E was excluded the effect of exogenous gibberellin. It is likely that the measured GA3 (sample G and A+G) was derived and moved from treated GA3 itself.
